# Direct Growth of Patterned Vertical Graphene Using Thermal Stress Mismatch between Barrier Layer and Substrate

**DOI:** 10.3390/nano13071242

**Published:** 2023-03-31

**Authors:** Fengsong Qian, Jun Deng, Xiaochen Ma, Guosheng Fu, Chen Xu

**Affiliations:** 1Key Laboratory of Optoelectronics Technology, Ministry of Education, Beijing University of Technology, Beijing 100124, China; 2Fert Beijing Institute, School of Microelectronics, Beihang University, Beijing 100191, China

**Keywords:** vertical graphene, direct growth, patterning, thermal stress mismatch, warpage, plastic deformation

## Abstract

Vertical graphene (VG) combines the excellent properties of conventional graphene with a unique vertical nanosheet structure, and has shown tremendous promise in the field of electronics and composites. However, its complex surface morphology brings great difficulties to micro-nano fabrication, especially regarding photolithography induced nanosheet collapse and remaining chemical residues. Here, we demonstrate an innovative method for directly growing patterned VG on a SiO_2_/Si substrate. A patterned Cr film was deposited on the substrate as a barrier layer. The VG was synthesized by PECVD on both the patterned Cr film and the exposed SiO_2_/Si substrate. During the cooling process, the patterned Cr film covered by VG naturally peeled off from the substrate due to the thermal stress mismatch, while the VG directly grown on the SiO_2_/Si substrate was remained. The temperature-dependent thermal stress distribution in each layer was analyzed using finite element simulations, and the separation mechanism of the Cr film from the substrate was explained. This method avoids the contamination and damage caused by the VG photolithography process. Our work is expected to provide a convenient and reliable solution for the manufacture of VG-based electronic devices.

## 1. Introduction

Vertical graphene (VG), also known as vertically oriented graphene, carbon/graphene nanowalls, or carbon/graphene nanosheets, consists of networks of graphene flakes arranged approximately perpendicular to a substrate [1]. The formation of VG is attributed to the vertical upward deposition of carbon reactive radicals (from plasma stimulation or thermal activation) at the crack edges of a the graphite base layer parallel to the substrate [2,3,4,5,6,7]. In this vertical growth mode, on the one hand, VG can be synthesized directly on non-metallic substrates such as Si, GaN, sapphire and quartz, without metal catalysts [5,8]. On the other hand, some unique morphologies are generated, including standing scattered layers, exposed sharp edges, and huge surface-to-volume ratios [1]. These characteristics give VG a wide range of potential applications in supercapacitors, photodetectors, bio- and gas-sensors, flexible electronic devices, etc. [9,10,11,12,13].

Although the development and research on VG has been carried out for many years, there is no ideal solution for the patterning technique of VG. The conventional photolithography etching process is not a suitable choice. The main reason for this is that these vertical nanosheets composed of VG, with a non-stacking morphology and a thickness of only a few atomic layers [3,14,15], are prone to tilt, bend and even be flattened during lithography. Furthermore, the large number of gaps between the vertical nanosheets also makes it more difficult to remove photoresists, which is a serious threat to the electrical properties of VG.

Initially, the excess VG on the substrate was usually stripped off with tape [16,17]. This simple and rudimentary method can protect the delicate nanosheets from damage and prevent them from contacting the photoresist. This method is very low cost and easy to operate, but it cannot support be used on patterns with complex geometries or precise dimensions. Recently, an ultrasonic etching technique using adhesion for patterning VG has been proposed [18]. Although photoresists are still used as a protective layer, unlike in the past, the VG is surface-modified by O_2_ plasma before spin-coating the photoresist to reduce the strong adhesion between the VG and the photoresist. After etching, the photoresist with opposite polarity to the VG is dissolved and removed cleanly by acetone. However, the treatment of O_2_ plasma inevitably affects the roughness and ion doping level of VG. Yu et al. have successfully achieved a direct patterned growth of VG by artificially designing an electric field distribution [19]. Under the electric-field-dominated growth mechanism, VG grows only on the area covered by the grounded gold electrodes on the substrate. Unfortunately, due to the uneven distribution of the space electric field, the growth in VG tends to exceed the expected area and the VG patterns become distorted. Therefore, a VG patterning process with less damage, lower contamination, and higher precision remains a challenge.

In this work, a feasible method for directly growing patterned VG is reported. Firstly, a layer of patterned Cr film was deposited on the SiO_2_/Si substrate as a barrier layer by sputtering. The synthesis of VG was then performed simultaneously on the surface of the Cr film and the exposed SiO_2_ (not covered by the Cr film) at 930 °C. During the cooling process at the end of the synthesis, the VG/Cr (the Cr film and the VG grown on it) cracked and warped due to thermal stress mismatch and thus separated automatically from the substrate. The VG directly grown on the SiO_2_ surface was remained. The self-separation mechanism of the VG/Cr from the substrate was clarified by finite element simulations of the stress evolution. This growth method achieved precisely patterned VG without debris residue and structural damage. We believe that this direct growth is an efficient and low-threshold solution for VG patterning and is expected to advance the preparation and application of VG and even other nanostructured.

## 2. Materials and Methods

### 2.1. Growth of Patterned VG

The growth procedure of the patterned VG is shown in Figure 1. In our experiments, VG was synthesized directly onto a Si substrate (Figure 1a) with a SiO_2_ layer (300 nm, dry thermal). A patterned barrier layer of a 500 nm thick metal Cr film was prepared on the substrate first to define the blank areas of VG through UV lithography, magnetron sputtering, and the lift-off technique sequentially. In our experiment, each of the four corners of the substrate were covered by a square shaped Cr film so that a “cross” shaped pattern of VG is present in the middle of the substrate (Figure 1b). The growth of VG (Figure 1c) was then carried out, following the process described in our previous report [11]. In detail, the sample (SiO_2_/Si substrate with patterned Cr film) was loaded into a PECVD system and heated to 930 °C at a rate of 300 °C/min in a H_2_ atmosphere (10 mbar) and then held for 20 min. The VG was grown for 40 min with a gas mixture of CH_4_:H_2_:Ar = 5:15:30 sccm, a plasma power of 30 W, and a chamber pressure of 5 mbar. After growth, the temperature was reduced to below 100 °C at a rate of 200 °C/min and the sample with VG was taken out from the PECVD system. At this moment, the VG grown directly on the exposed SiO_2_ surface was still firmly attached to the substrate, while the VG/Cr in the defined blank area had cracked and warped, and automatically separated from the substrate (Figure 1d). Finally, the residue of the VG/Cr debris on the substrate was removed by N_2_ purging (Figure 1e), and patterned VG (Figure 1f), present only outside the blank area, was produced.

### 2.2. Characterization of Patterned VG

The quality and distribution of the patterned VG were clarified by Raman spectroscopy (LabRAM HR, Horiba, Kyoto-shi, Japan, 532 nm laser source).The morphology of the patterned VG on the front and sides was characterized using scanning electron microscopy (SEM; Merlin, Zeiss, Oberkochen, Germany, 10 kV) and atomic force microscopy (AFM; Dimension XR, Bruker, Billerica, MA, USA).

### 2.3. Finite Element Simulation

The distribution and evolution of thermal stress in the sample from heating to cooling were simulated by the finite element software ANSYS. A two-dimensional (2D) planar shear stress model was considered. The mesh models used for finite element analysis are shown in Appendix A. In the models, the lengths of the Cr pattern and the SiO_2_/Si substrate were set to 4.5 μm and 6 μm, respectively. The layer thicknesses of VG, Cr and SiO_2_ were set to the actual values. The thickness of the Si was set to 1.5 μm instead of the actual value. This is because the Si substrate is too thick compared with other layers, and using the actual size is not beneficial for highlighting the stress differences at the interface. Other input parameters for the model, including: thermal conductivity, CTE, elastic modulus and Poisson’s ratio, are shown in Appendix A. Since the physical properties of VG have not been reported, the parameters of carbon nanotubes and pyrolytic graphite are used here as a substitute.

## 3. Results and Discussion

In our experiment, a “cross” VG pattern was designed and a patterned Cr film was deposited on the area outside the “cross”, as shown in Figure 2a. The purpose of preparing this layer of Cr film was to prevent the VG from growing on the substrate area covered by the Cr. In addition, metal Cr with a high melting point (about 1907 °C [20]) was chosen to ensure the integrity and regularity of the pattern, since it would not melt or agglomerate during the synthesis of VG. The synthesis of VG can be performed indiscriminately on the surfaces of the Cr and the exposed SiO_2_. The reason for this is that, the reactive carbon intermediates decompose from CH_4_ molecules at the initial stage of VG growth, achieving fast surface carbonization and resulting in the formation of a graphitic buffer layer on the substrate through intense ion bombardment under PECVD conditions [21,22], rather than requiring metal-assisted catalysis as in conventional planar 2D graphene films [8,23]. Figure 2b shows the appearance of the sample taken from the PECVD. On the “cross” area, the VG flatly covers on the SiO_2_/Si substrate. Outside the “cross” area, most of the VG/Cr has fallen off. Only a small amount of VG/Cr debris left on the substrate is warped. This residual VG/Cr debris can be easily blown off by N_2_ purging, as shown in Figure 2c. In this case, the barrier layer was completely removed without additional etching treatment. The lithography-free and etch-free patterned growth method is simple and convenient, and can effectively avoid organic contamination.

The quality of the grown VG and the effect of patterning were examined by Raman spectroscopy. We compared the measurement results on the “cross” area (red line) and outside the “cross” area (black line) in Figure 3a. In the “cross” area, the Raman spectrum contained three main bands: a D band (~1350 cm^−1^), a G band (~1580 cm^−1^), and a 2D band (~2690 cm^−1^), which are typical bands of graphene [24]. The intensity ratio of the D band to the G band (I_D_/I_G_) indicates the defect degree of the graphene nanosheets in the VG, and the intensity ratio of the 2D band to the G band (I_2D_/I_G_) reflects the number of layers in the graphene nanosheets [15,24]. The VG has an I_D_/I_G_ value of about 0.71, which indicates a high defect state. This may be caused by the large number of edges and domain boundaries of the graphene nanosheets. The I_2D_/I_G_ value is about 0.64, which indicates a low-layer structure of the nanosheets [15,25,26]. Outside the “cross” area, no graphene-related bands were found, suggesting the absence of VG here.

Figure 3b–d show the Raman mapping images of the D, G, and 2D bands of the patterned VG. It can be observed that the Raman signal is precisely distributed only within the “cross” area and the signal intensity is quite uniform, which demonstrates the uniformity of the grown VG and the high precision of the pattern. Moreover, it also proves that the warping of VG/Cr occurs after the growth of VG. If the warping occurs before or during the growth of VG, VG will also be generated on the SiO_2_ surface area originally covered by the Cr film, and a Raman signal in this area will be detected. The Raman mapping of I_D_/I_G_ and I_2D_/I_G_ of the VG shown in Figure 3e,f, is generally consistent with the calculated results from the single spectrum in Figure 3a. The defect density ($n_{D}$) and average domain size (La) of the VG could be quantified based on the I_D_/I_G_ value according to the following equations [27,28]:(1)$$n_{D}=\left(1.8\pm 0.5\right)\times 10^{22}\cdot {\lambda_{L}}^{-4}\left(I_{D}/I_{G}\right)$$
(2)$$La=2.4\times 10^{-10}\cdot {\lambda_{L}}^{4}{\left(I_{D}/I_{G}\right)}^{-1}$$
where $\lambda_{L}$ is the wavelength of the laser used for Raman measurement. The defect density of the VG reaches approximately 1.6 × 10^11^ cm^−1^. The average domain size is about 27 nm, a value that is at the same level as that of previously reported VG [14].

The patterned VG was analyzed by SEM and AFM characterizations. As shown in Figure 4a, the VG pattern is complete and precise, and the edges are neat. The surface of the sample is quite clean and free of VG/Cr debris, which reveals that N_2_ purging is effective for removing VG/Cr residues. Here, we also note that the surface of SiO_2_ is flat and intact, which indicates that VG/Cr shedding did not cause mechanical damage to the SiO_2_/Si substrate. Figure 4b,c shows the morphology of the side and front of this VG. From the side, these graphene nanosheets exhibit an overall translucent appearance and roughly vertical orientation to the substrate. From the front, the nanosheets show highly random lamellar and sheet-like features. These are typical morphologies of VG [8,29]. The large number of nanosheet edges presented in Figure 4c corroborates the relatively high D band shown in Figure 3a,e.

We performed conventional VG patterning using a photolithography-O_2_ plasma etching process to contrast with this direct patterned growth. Due to the extrusion from the photoresist, the graphene nanosheets are in the state of collapse and lateral stacking, as shown in Figure 4d, which is quite unfavorable for the applications of VG. The vertical orientation, exposed sharp edges, and large surface-to-volume ratio are important characteristics of VG. For example, when VG is applied in photodetection, the larger light-receiving surface and the multiple light reflections between vertical nanosheets can greatly improve the light absorption [23]. However, the irrecoverable damage brought by photolithography will undoubtedly weaken the performance of VG in practical applications. In comparison, our direct patterned growth bypasses the photolithography-etching process, which allows the original VG morphology to be preserved. According to the AFM measurement results shown in Figure 4e, the height of the grown VG is 850 nm. The 3D AFM image in Figure 4f further proves that the VG has a vertical orientation, discrete structure and ultra-high surface ratio.

The core advantage of our patterned growth method is that the VG/Cr can be naturally separated from the substrate, which is thought to be due to the mismatch in thermal stress between the Cr film and the SiO_2_/Si substrate. Due to differences in physical properties (mainly the coefficient of thermal expansion, CTE), stress tends to be induced in the film-substrate complex during heating or cooling processes, which is known as thermal stress [30,31,32]. When the temperature reaches a certain point (yield temperature), the film is unable to withstand the intense thermal stress (yield stress) resulting from the great change in temperature and is subjected to plastic deformation, including: cracking, warping and delamination [30,31,33]. In the fields of materials science and microelectronics, thermal stress is commonly a problem that cannot be ignored, because it can damage the original structure of devices or materials and thus degrade the electrical properties [34,35,36]. Only in a few cases, does thermal stress play a positive role, e.g., Ohnishi has reported the use of thermal stress at heterogeneous interfaces to achieve self-separation of GaN films from substrates, and pointed out two factors that drive this self-separation: two shear stresses in the opposite directions generated within the film and the substrate during cooling, and cracking caused by the maximum shear stress appearing at the sidewalls near the interface [37]. This self-separation mechanism is also applied in this work. The difference is that, on the one hand, the VG/Cr film was separated from the substrate under the action of thermal stress. On the other hand, since the CTE of VG is similar to that of SiO_2_/Si, the VG directly grown on the SiO_2_ layer was not deformed. Finally, a selective growth was realized. Typical non-metallic substrate materials, such as GaN, SiC, sapphire, and quartz, generally have lower CTEs than SiO_2_/Si; therefore, it can be predicted that our patterned growth method is equally applicable on these substrates.

In order to verify this self-separation mechanism, the variation in thermal stress with temperature was simulated. The temperature change in the sample throughout the VG growth process can be divided into three stages. In stage I, the substrate with the patterned Cr film is heated in PECVD system from room temperature (25 °C) to growth temperature (930 °C). In stage II, H_2_ annealing and VG growth are successively performed at 930 °C. In stage III, the temperature drops from 930 °C to the unloading temperature (100 °C), where the VG/Cr cracks and warps at the yield temperature. In addition to the temperature covariate, we also considered the stress relaxation mechanism at high temperatures. That is, high temperature annealing can promote the diffusion of atoms and grain boundaries in deposited metal film, eliminate defects, and thus reduce the thermal stress [38,39]. In our experiments, stage II has the longest duration and the highest temperature, and the thermal stress relaxation is mainly concentrated in this stage. Here we simplify the model and assume that the thermal stresses in the Cr film are completely removed (drop to 0) in stage II. Four reference paths were selected to investigate the stress distribution at different temperatures. Figure 5a–d shows the stress distribution of the SiO_2_/Si substrate with patterned Cr film ramped up to 300 °C and 600 °C, and the SiO_2_/Si substrate with VG and patterned Cr film ramped down to 600 °C and 300 °C, respectively. In the current work, a positive value indicates tensile stress and a negative value indicates compressive stress, and it is assumed that the VG/Cr has not yet deformed when the temperature drops to 300 °C.

During the heating process, the Cr film is subjected to compressive stress while the SiO_2_ layer is subjected to tensile stresses at the Cr-SiO_2_ interface. This is mainly due to the fact that the CTE of Cr is much larger than that of SiO_2_ (Appendix A), and the tendency of the Cr film to expand is higher and limited by the SiO_2_ layer with a relatively weak expansion tendency. As the temperature increases, the difference between the physical parameters of Cr and SiO_2_ increases further compared to that before heating, so the compressive stress in the Cr film and the tensile stress in the SiO_2_ layer also increase correspondingly. In stage II, the stress in the Cr film is defined as 0 due to the thermal stress relaxation mechanism. In the following cooling process, the temperature reduction causes the Cr film, the SiO_2_ layer and the introduced VG film to contract. Since the CTE of Cr is larger than that of both VG and SiO_2_, the contraction of the Cr film is larger and limited by the VG film and SiO_2_ layer with relatively weaker contraction tendencies. As a result, the stress in the Cr film changes from compressive stress during the heating process to tensile stress, the stress direction in SiO_2_ layer also changes, and the VG grown on the Cr film is subjected to the same compressive stresses as the SiO_2_ layer. These stresses become larger and larger as the temperature decreases. Since the CTEs of VG, SiO_2_ and Si are similar, the difference in thermal stress between the VG–SiO_2_ and SiO_2_–Si interfaces is smaller than that between the VG–Cr and Cr–SiO_2_ interfaces, which could partially explain why VG/Cr was separated from the substrate, while VG grown directly on the substrate was not.

As an important research object, the equivalent stress transfer curves in the Cr film were extracted, as shown in Figure 6. The Cr film always bears compressive stress during heating, and its absolute value becomes larger with increasing temperature. When the temperature rises to 930 °C, the compressive stress reaches a maximum (σ_max_= −660 MPa). During the cooling period, tensile stress is applied to the Cr film, which gradually increases with decreasing temperature. The stress transfer rate at the cooling stage is generally higher than that at the heating stage, because the Cr film at the cooling stage additionally experiences tensile stress from the grown VG. The point where the temperature dropped to 280 °C is deliberately marked, and the stress value here is the same as the maximum compressive stress during heating. The deformation of the chromium film is considered to be related only to the magnitude of the stress and not to the direction, it can be roughly inferred that the cracking and warping of VG/Cr occurred below 280 °C.

Based on the above simulation results, we clarified the mechanism of this self-separation and patterned growth. As shown in Figure 7a, the Cr film is sputtered onto the SiO_2_/Si substrate as a barrier layer prior to heating, which has a pattern opposite to the VG to be grown and serves to block the direct growth of the VG on the SiO_2_ surface covered by the Cr film. Since the sample is at room temperature, the thermal stress effect inside the sample can be ignored. During heating, compressive and tensile stresses are applied in the Cr film and the underlying SiO_2_/Si substrate, respectively. These stresses increase with temperature and reach maximums at the temperature of synthesis (930 °C), as shown in Figure 7b,c. At this stage, the stresses applied in the Cr film do not reach the yield stress, so the Cr film does not deform. During the subsequent application of a constant temperature of 930 °C, the Cr film is not subjected to thermal stresses due to the high temperature stress relaxation mechanism, and the interiors of the grown VG and the SiO_2_/Si are also considered to be free of thermal stresses, as shown in Figure 7d. During cooling, the Cr film is subjected to greater stress because of the introduction of the VG. When the temperature drops to the yield point (T_y_ < 220 °C), the VG/Cr cracks and warps, and is then automatically separated from the substrate, as shown in Figure 7e,f. We found that delamination occurs only at the interface between the Cr film and the SiO_2_ layer, although stress is present throughout the Cr body, which may be caused by the weak adhesion of the sputtered Cr to the thermally oxidized SiO_2_ surface [38,39]. Besides that, it is possible that some carbon atoms diffuse into the Cr side to form a buffer layer during VG growth [40]. Therefore, delamination occurs at the Cr–SiO_2_ interface rather than the VG–Cr interface. The result is that the VG grown on the Cr barrier layer falls off with the Cr barrier layer, while the VG grown directly on the SiO_2_ layer contacts the substrate firmly and forms a layout contrary to the Cr pattern.

Currently, our direct growth method of patterned VG can cover most of the requirements (high patterning precision, high VG quality, no-contamination and convenient operation), but there is still one shortcoming. According to our experimental results, this method is only applicable to the growth of VG with a height of more than 200 nm. Below this value, the VG will not be able to provide enough stress to the Cr film during cooling, and in turn, the VG/Cr will not smoothly separate from the substrate. For this problem, some optimization schemes can be adopted in the future, such as appropriately increasing the thickness of the Cr film [37,41], or selecting materials with a suitable CTE as barrier layers.

## 4. Conclusions

In summary, we achieved a direct patterned growth of VG on a substrate. A sputtered Cr film is used as a barrier layer, which can prevent the synthesis of VG on unwanted areas and separate automatically from the substrate due to thermal stress mismatch during the cooling process after the growth of VG. Raman measurements show that patterned VG with a consistent growth quality (I_D_/I_G_ ≈ 0.71 and I_2D_/I_G_ ≈ 0.64) was distributed only in the expected area. The morphology characterization results confirm that the patterned VG prepared by this method has high precision, a clean surface, neat edges, and an undamaged morphological structure. The evolution of the internal stresses throughout the VG growth process is described by finite element simulations and the self-separation mechanism is revealed. Due to being covered by VG, the Cr film is subjected to stronger stresses during cooling than during heating. With the decrease in temperature, the stress gradually increases and eventually reaches the yield stress, at which point the Cr film undergoes plastic deformation. From the calculated values of the stresses in the Cr film, it is speculated that the self-separation occurs when the temperature drops below 220 °C. This patterned growth method provides a promising path for VG-based electronic device fabrication, and inspires the selective preparation of other vertically oriented nanomaterials, such as GaN nanowalls, Mn_2_O_3_ nanowalls and MgZnO nanowalls.

## Figures and Tables

**Figure 1 nanomaterials-13-01242-f001:**
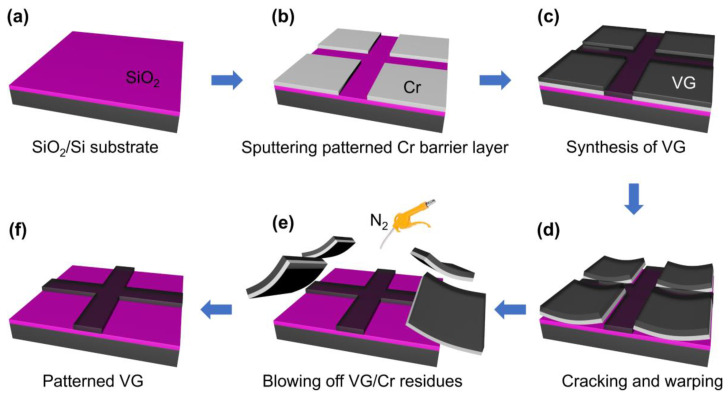
Schematic illustration of the growth process for the patterned VG. (**a**) SiO_2_/Si substrate on which the patterned VG was grown. (**b**) Sputtered Cr barrier layer was deposited on the substrate to define the blank area of VG. (**c**) VG was synthesized by PECVD. (**d**) The VG/Cr cracked and warped, and then separated from the substrate during the cooling process. (**e**) Residual VG/Cr debris left on the substrate was blown off by N_2_. (**f**) The patterned VG was generated.

**Figure 2 nanomaterials-13-01242-f002:**
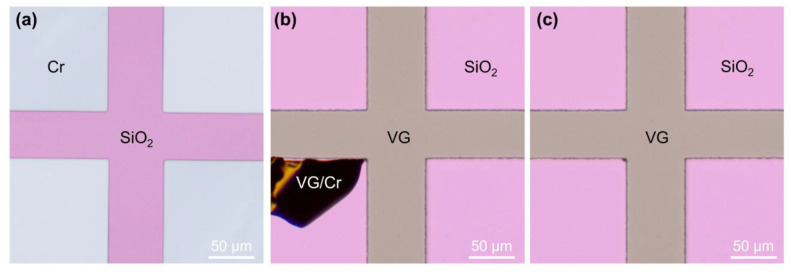
Microscopic images of (**a**) patterned Cr barrier layer, (**b**) warped VG/Cr debris left on the substrate, and (**c**) final patterned VG.

**Figure 3 nanomaterials-13-01242-f003:**
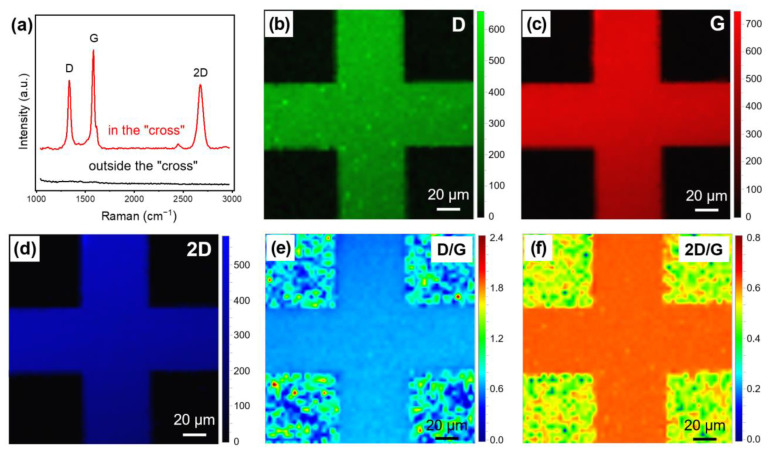
Raman measurements of the patterned VG grown directly on a SiO_2_/Si substrate. (**a**) Raman spectra of random points on and outside the patterned VG area. (**b**–**f**) Raman mapping images of (**b**) I_D_, (**c**) I_G_, (**d**) I_2D_, (**e**) I_D_/I_G_, and (**f**) I_2D_/I_G_. The mapping area is 150 μm × 150 μm, and 2601 data points were used.

**Figure 4 nanomaterials-13-01242-f004:**
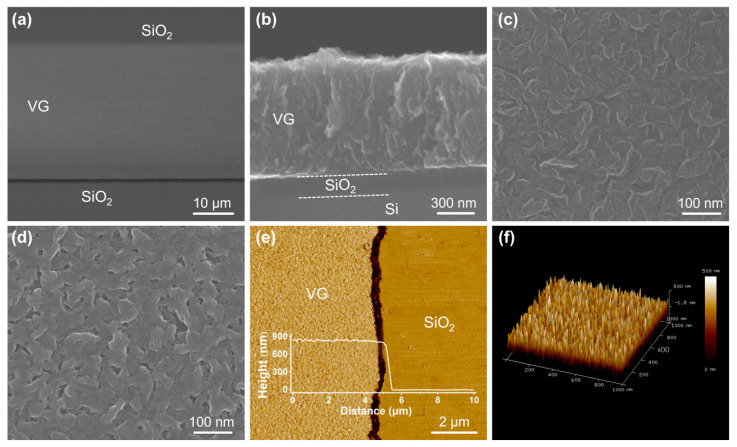
Morphology of the patterned VG: (**a**) oblique view SEM image, (**b**) cross-sectional view SEM image, and (**c**) top view SEM image of the patterned VG grown directly on a SiO_2_/Si substrate. (**d**) Top view SEM image of the patterned VG prepared using conventional photolithography-O_2_ plasma etching process. (**e**) Two-dimensional image of the grown VG pattern boundary in a 10 μm × 10 μm area. (**f**) Three-dimensional AFM image of the grown VG distribution in a 1 μm × 1 μm area.

**Figure 5 nanomaterials-13-01242-f005:**
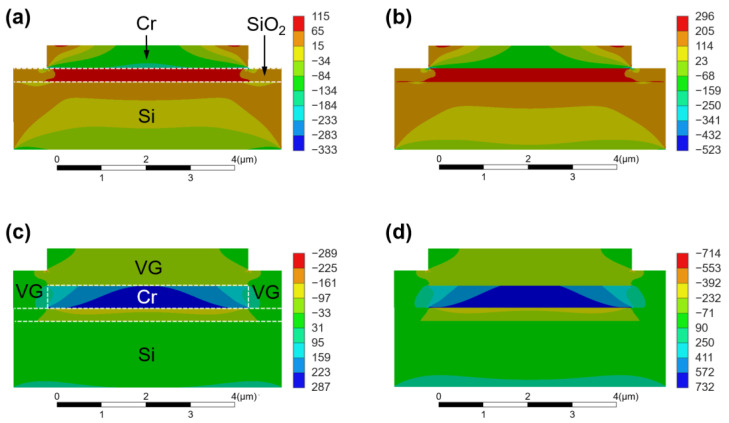
Stress (in MPa) distributions in the SiO_2_/Si substrate with patterned Cr film at (**a**) 300 °C and (**b**) 600 °C during heating before synthesis of VG. Stress (in MPa) distributions in the SiO_2_/Si substrate with VG and patterned Cr film at (**c**) 600 °C and (**d**) 300 °C during cooling after synthesis of VG.

**Figure 6 nanomaterials-13-01242-f006:**
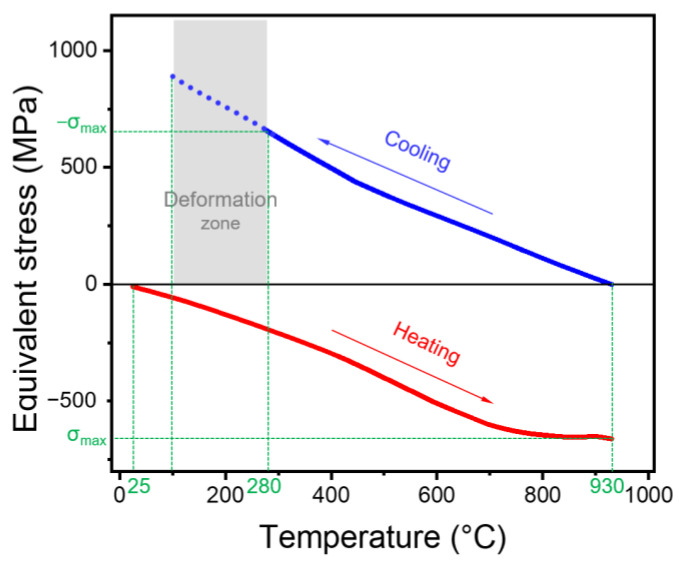
Simulation-based equivalent stress applied in the Cr film as a function of temperature during heating and cooling processes.

**Figure 7 nanomaterials-13-01242-f007:**
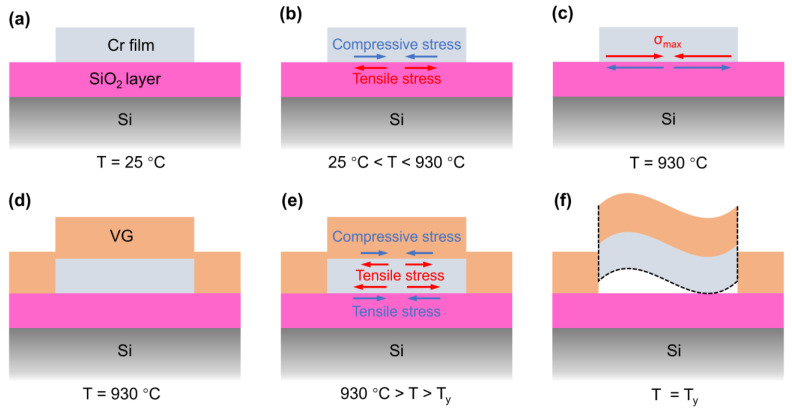
Schematic of patterned VG growth mechanism: (**a**) Before heating and growth of VG, there is almost no stress inside the SiO_2_/Si substrate with patterned Cr film. (**b**) During heating, the absolute values of the compressive stress (blue line) applied in the Cr film and the tensile stress (red line) applied in SiO_2_/Si increase with the increase in temperature. (**c**) The compressive stress in the Cr film reaches a maximum (σ_max_ = −660 MPa) when the temperature rises to 930 °C. (**d**) During the VG growth period, the stresses in the sample are relaxed. (**e**) During cooling, tensile stress is applied in the Cr film, and compressive stress is applied in both the VG film and SiO_2_/Si, both of which increase with the decrease in temperature. (**f**) When the temperature drops to T_y_, VG/Cr cracks and warps, and then automatically separates from the substrate.

## Data Availability

All data can be obtained from the corresponding author.

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
