# Peer review of "Direct Growth of Patterned Vertical Graphene Using Thermal Stress Mismatch between Barrier Layer and Substrate"

_nanomaterials, 2023, doi:10.3390/nano13071242_

Round 1

Reviewer 1 Report

See the attachment

Reviewer 2 Report

On the whole, the article is quite original and fits the content of the journal.
I have some doubts about the correct choice of name and terminology.
The authors talk about vertical graphene; however, the material they obtain is
highly defective and does not at all look like a two-dimensional material;
is not graphene in any way and does not exhibit its characteristic properties.
Apparently, these films are amorphous carbon. In my opinion, the authors should
replace “vertical graphene” with “amorphous carbon” everywhere in the text
of the article and the title, after which the article can be published.

Reviewer 3 Report

The authors reported a new approach to prepare patterned vertical graphene by utilizing thermal stress mismatch. The work is of value and suits the journal. My main concern is, the scale of PVG growth is rather small, does the FEM model apply at this scale? A verfication/validation of model is required. After model validation, the mechanism would be sound. 

Round 2

Reviewer 2 Report

The authors convinced me that the term "vertical graphene"
had previously been used and applied
to similar materials as investigated in this article.
But I remained in my opinion that this is an unfortunate term.
Reading it, people might think
that we are talking about the study of two-dimensional material,
but this is not so. The article can be published in its current form.